# How to Improve the Acceptance of Autonomous Driving Technology: Effective Elements Identified on the Basis of the Kano Model

**Jong-Gyu Shin [1]** , **In-Seok Heo [2]** , **Jin-Hae Yae [3]** and **Sang-Ho Kim [2],\***

[1] Regional Industrial Management Research Institute, Kumoh National Institute of Technology, Gumi 39177, Korea; shinjg@kumoh.ac.kr

[2] Department of Industrial Engineering, Kumoh National Institute of Technology, Gumi 39177, Korea; his4113@kumoh.ac.kr

[3] Korea Automotive Technology Institute (Katech), Cheonan 31214, Korea; jhyae@katech.re.kr

**\*** Correspondence: kimsh@kumoh.ac.kr; Tel.: +82-054-478-7656

**Abstract:** Innovative sociotechnical change is forthcoming because of autonomous driving; however, only a few studies have focused on the acceptance of this technology, which is not up to social expectation. In this study, we present and validate a research framework on the basis of the Kano model to identify the effective acceptance elements for autonomous driving technology. By collecting and analyzing the survey data of 187 people, it was confirmed that the elements of acceptance for autonomous driving technology can be classified according to the Kano attributes. This means that these acceptance elements should be resolved with priority in order to secure the acceptance. Legal policies and ethical guidelines are identified as top priorities for ensuring the acceptance of autonomous driving. Traffic congestion, situational awareness, malfunction prevention, and fatigue/stress relief must be addressed as utmost priorities. The framework and results from this study can be used to establish efficient strategies for developing autonomous driving technologies according to the user requirement levels.

**Keywords:** Kano model; autonomous driving technology; human factor; user acceptance; cross-sectional study

## 1. Introduction

The Industrial Revolution was an outcome of continuous technological development, and it profoundly improved the quality of people's lives, and brought paradigmatic shifts in various fields, including industry, culture, and the economy. Human requirements continue to evolve with the improving quality of human life as time and the sociotechnological environment progress [1]. Likewise, technology develops because it coevolves with human values (Figure 1) [2]. Developers must understand the user requirements of a period and establish efficient research and development strategies on the basis of that understanding [3]. If mismatches arise between users and developers with regard to the acceptance of a technology, challenges can ensue when a new technology is introduced [4]. It is important to note that any attractive technological characteristic cannot maintain its utility for user satisfaction indefinitely. These days, user values have progressed to include usability and good user experiences, which are far beyond the functionality and reliability values of a technology [5]. It should also be noted that people differ, and the level of user acceptance of a technology can likewise differ. Various factors determine technological acceptance, and their effects are not homogeneous for all users [6].

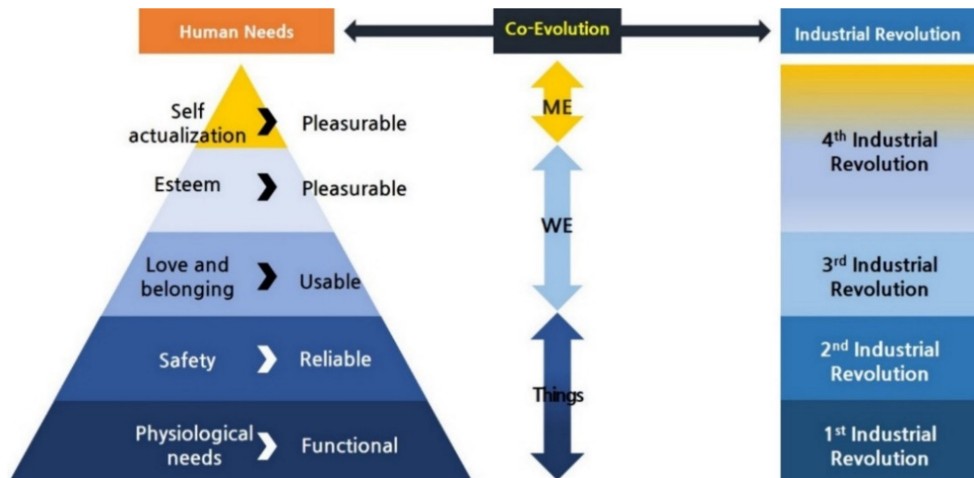

**Figure 1.** Relationship between hierarchy of human needs and the evolution of the Industrial Revolution.

Along with artificial intelligence and Internet of Things technologies, autonomous driving is representative of the Fourth Industrial Revolution. Its commercialization is being accelerated by the rapid developments in information and communication technology [7]. Research and developments in autonomous driving are supported at various national levels, and several automobile companies are rapidly developing the relevant technologies [8]. However, according to research conducted by the Korea Research Institute for Human Settlements (KRIHS), most of these studies are only focused on developing the functional aspects of the technology [9]. Although studies related to transportation systems are being conducted, their discussions of the social impacts are relatively insufficient [10]. It is noted that the specific definitions for: (a) the factors determining the acceptance of autonomous driving; and (b) the systematic investigation methods, are also unclear [10]. "The higher the functional performance and reliability, the better the user acceptance" seems to be a faulty assumption that is commonly found in technological acceptance studies. As mentioned previously, the needs of users are changing and the contribution of a specific factor on technological acceptance becomes less critical when that factor is more atypical [11]. To solve the shortcomings of previous research, we must identify the criteria for classifying components that affect the acceptance of autonomous driving technology; moreover, analysis and interpretation are required to provide specific information regarding the driver acceptance of autonomous driving.

The Kano model classifies the quality attributes of a product or service into several factors, according to the customer satisfaction and product function [12]. In addition, it is excellent at deriving the direction of product development because it accommodates the needs of customer satisfaction and interests [13–15]. A Kano model implementing these features is considered to be an appropriate research method to compensate for the shortcomings of previous research because it can capture the varying acceptances of a technology across various dimensions. It is also considered to be effective for understanding the differences in user requirements with regard to technology [16]. In this study, we use the Kano model as a tool to identify and categorize the factors determining the acceptance of autonomous driving technology. A research framework based on this model is presented, and its validity is reviewed. The proposed framework can help to establish the foundations for research methodologies that facilitate social countermeasures and technology development strategiesto improve the acceptance of autonomous driving technologies.

## 2. Background

### 2.1. Autonomous Driving Technology

Automobiles are designed with the aims of increasing their efficiency of mobility and ensuring the safety of the driver. Research is being actively conducted into the development of intelligent vehicles that are equipped with various driving assistance functions that can reduce the user's driving burden [8]. The study of advanced driver assistance systems is based on the premise that the primary cause of traffic accidents is driver carelessness; moreover, inefficiency in driving is also attributed to drivers. "Autonomous driving" means that a car drives to a destination by recognizing the surrounding environment, assessing the driving situation, and controlling the vehicle itself, without driver intervention [17]. Autonomous vehicles are emerging as a future means of personal transportation that can reduce traffic accidents, increase traffic efficiency, save fuel, and increase driver convenience [18].

Autonomous driving technology can be classified into stages, according to the level and role of a function. The Society of Automotive Engineers classifies autonomous driving technologies into six levels (0–5), depending on the steering control, speed adjustment, driving controller, driving situation, and environmental monitoring agents [19]. Level 0 (no automation) refers to complete manual driving, without any driving assistance functions. Level 1 (driver assistance) automation is equipped with certain auxiliary functions that help the driver; this is the most commonly used level of driving assistance technology. Level 2 (partial automation) requires users to look ahead, though the system can perform the main control functions, such as steering and speed adjustment. From Level 3 (conditional automation) upwards, the system monitors the driving environment; although the system performs the main control functions, an emergency response must be inputted by the driver, and the responsibility lies with them. From Level 4 (high automation), no user intervention is required, and the system performs control functions, monitors the driving environment, and handles emergency situations at a high level of automation; however, such autonomous driving is possible only within a specific area, where it is fully learned. In Level 5 (full automation), the autonomous driving system can control all the autonomous driving systems without a driver, and it can autonomously drive to a location without restrictions. Therefore, it is important to identify the technical issues related to the user acceptance in the development of autonomous driving technology to ensure that they evolve simultaneously.

### 2.2. Kano Model

The Kano model, developed in 1984, is based on Herzberg's motivation-hygiene theory (1968) and the law of diminishing marginal utility. It considers the subjective aspects of satisfaction and dissatisfaction, and the objective aspect of functionality [20]. The role of quality can be classified into five categories, according to the degree with which acceptance and satisfaction are provided to users, and using a classification form based on the two dimensions of the Kano model:

1. Attractive quality attributes (A): A sufficient conditional quality attribute that provides satisfaction when achieved fully, but that does not cause dissatisfaction when unfulfilled;
2. One-dimensional quality attributes (O): A necessary conditional quality attribute that results in satisfaction when fulfilled, and dissatisfaction when unfulfilled;
3. Must-be quality attributes (M): A necessary attribute that, if not fulfilled, will cause customer dissatisfaction; however, if it reaches a certain level, the customer is no longer satisfied;
4. Indifferent quality attributes (I): A quality attribute that does not significantly affect customer satisfaction, regardless of whether it is fulfilled;
5. Reverse quality attributes (R): A quality attribute that, if fulfilled, causes dissatisfaction and, if unfulfilled, it satisfies the customer.

The attributes in the Kano model can be primarily classified using questionnaires; an exemplary questionnaire is shown in Table 1 below.

**Table 1.** Kano model questionnaire type.

| | Sample Question | | Answer |
|---|---|---|---|
| Functional form of the question | What if autonomous driving **could** **effectively** complement your driving skills? | → | I like it that way |
| | | → | It must be that way |
| | | → | I am neutral |
| | | → | I can live with it that way |
| | | → | I dislike it that way |
| Dysfunctional form of the question | What if your autonomous driving skills are **not as effective as** your driving skills? | → | I like it that way |
| | | → | It must be that way |
| | | → | I am neutral |
| | | → | I can live with it that way |
| | | → | I dislike it that way |

The Kano questionnaire presents a pair of questions (a functional form and a dysfunctional form) for an acceptance element, and the acceptance element is classified as one of the five dimensions, according to the answers. The classification criteria are summarized in Table 2 below.

**Table 2.** Kano model evaluation table.

| | | Dysfunctional form of the Question | | | | |
|---|---|---|---|---|---|---|
| | | Like | Must-Be | Neutral | Live With | Dislike |
| Functional form of the question | Like | Q | A | A | A | O |
| | Must-be | R | I | I | I | M |
| | Neutral | R | I | I | I | M |
| | Live with | R | I | I | I | M |
| | Dislike | R | R | R | R | Q |

A: Attractive; O: One-dimensional; M: Must-be; I: Indifferent; R: Reverse; Q: Questionable.

However, because the quality attribute classification under the Kano model is judged to be the most frequent quality attribute, relatively weaker quality attributes are ignored. To address these issues, the customer satisfaction coefficient presented by Timko (1993) was used to calculate the impact of the customer satisfaction and dissatisfaction [21]. The customer satisfaction coefficient determines the degree of user satisfaction or dissatisfaction when a user uses a product or service; it can be derived by calculating the frequencies corresponding to the following quality attributes:

$$\text{SI : Satisfaction Index} = \frac{A + O}{A + O + M + I} \tag{1}$$

$$\text{DI : Dissatisfaction Index} = (-1)\left(\frac{O + M}{A + O + M + I}\right) \tag{2}$$

The factors perceived to be more important can be predicted; then, from these, it is advantageous to classify the quality attributes into the necessary conditions (e.g., competitive advantage and indifference) for each attribute, and establish development strategies (Figure 2).

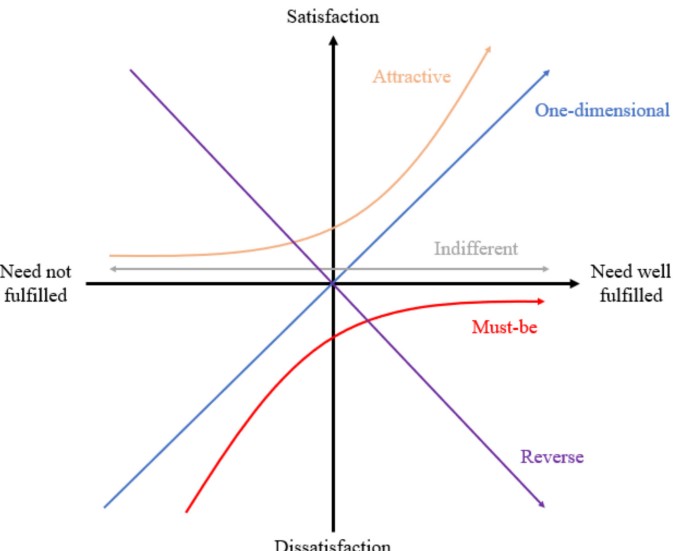

**Figure 2.** Kano model diagram.

The Kano model has been applied in various studies to determine the elements that affect user acceptance or satisfaction. Shin et al. used the Kano model to ensure user satisfaction with a function derived as a concept in the development of smart audio [22]. Kim applied the Kano model to evaluate the service quality of sports programs for the elderly [23]. Chen et al. applied it to evaluate the quality of transportation services [24]; there, elements closely related to the quality of transportation services for various tourists participating in a festival were evaluated. In this study, we attempt to determine and classify the elements that influence the customer acceptance of autonomous driving technologies.

## 3. Method

### 3.1. Acceptance Evaluation Framework Based on Kano Model

In this study, the attributes of the Kano model were reinterpreted according to the user's requirement steps. To classify the factors to the level that state-of-the-art autonomous driving technologies require, we applied the hierarchical structures of the user requirements provided by Walter [5] to classify the Kano category characteristics according to the user requirements. In terms of the functional and reliable levels of autonomous driving technology, R&D has been steadily progressing [25,26]. When comparing the timing of the introduction and the commercialization level of autonomous driving technology, user needs relevant to the functional level were classified as Must-be attributes, and those relevant to the reliable level were classified as Must-be or One-dimensional attributes. The usable aspects of user requirements, which relate to the interactions with users, are undergoing accelerated R&D in terms of the user interface/user experience designs (e.g., human–machine interfaces) in vehicles [27,28], and it is emerging as a key competitive edge of autonomous driving technology. The usable aspect is a necessary level of user needs and is currently a key component of autonomous driving technology; thus, it is classified under One-dimensional or Attractive attributes. The pleasurable level, the last stage of human needs, has the characteristics of user emotions (e.g., giving pleasure or improving user satisfaction). In the era of the Fourth Industrial Revolution, this can be seen as a future task to meet once the usability has been satisfied. Therefore, the pleasurable level, which relates to the driver's emotions regarding autonomous driving technology, was classified as an Attractive or Indifferent attribute because the human needs were judged to have not yet been met.

We propose a framework for researching the technological acceptance of autonomous driving, by applying the Kano model and the human requirement steps, and our research hypothesis is as follows:

**Hypothesis 1 (H1).** *The technological acceptance of autonomous driving is determined by various elements, each of which belongs to one of the five Kano categories. This means that the elements have different impacts on technological acceptance, and the differences can be characterized by the Kano categories.*

The model and framework for this study (based on the hypothesis) are depicted in Figure 3.

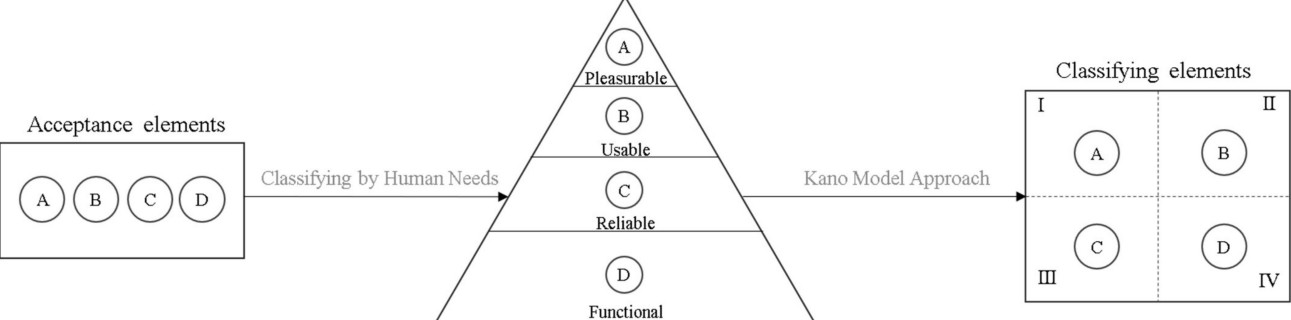

**Figure 3.** Study framework derived from the Kano model.

To apply the Kano model, we derived the potential elements that might affect the acceptance of autonomous driving technology. For the potential acceptance elements, the factors primarily used in previous studies were reviewed, and the factors with relatively meaningful results were derived. A questionnaire was conducted according to these potential acceptance elements. On the basis of the results, the acceptance elements for autonomous driving were classified. The validity of the Kano-model-based acceptance research framework was reviewed by identifying the acceptance elements for the user requirement step in each Kano category.

### 3.2. Identification of Acceptance Elements

To create two-dimensional questionnaires based on the Kano model, we first defined the potential acceptance elements that might affect the acceptance of autonomous driving. We reviewed previous studies to screen for those potential elements that could yield significant effects. Because mismatches arose in the definitions and ranges of each study, the potential acceptance elements to be considered were derived and classified as widely as possible. Research that has analyzed the subjective opinions of drivers with regard to autonomous driving was reviewed and collected. We refer to studies relating to the preference for and rejection of a new technology, including (but not limited to) autonomous driving. According to Nielson, system acceptance is classified as a social acceptance, which is related to social issues, such as regulations, institutions, and expectations, as well as a practical acceptance, which involves the detailed functional aspects of cost and technology [29]. Therefore, in this study, the collected acceptance elements were grouped into social and practical acceptance elements, as is summarized in Table 3.

Schoettle and Sivak conducted a survey of 1,533 adults over the age of 18 in order to identify their recognition of and intention to actually use autonomous vehicles [30]. Safety issues regarding malfunctions and system failures were found to be the biggest concern, along with vehicle movement whilst unoccupied. Jardim et al. conducted a survey that considered safety, cost, and legislation as the major elements, alongside energy efficiency, productivity, and the environment as secondary elements influencing the acceptance of autonomous driving technology [31]. Bansal et al. (2016) attempted to predict the changes in the application of connected automated vehicle technology with respect to various price scenarios in order to analyze the acceptance of autonomous driving technology [32]. Kim et al. analyzed the ethical issues regarding the intransigent behaviors that might arise

during autonomous driving, and they examined the social opinions regarding its legal liability, insurance, and advantages and disadvantages [33].

**Table 3.** Acceptance elements in autonomous driving technology.

| User Requirement | Practical Acceptance Elements | | Social Acceptance Elements | |
|---|---|---|---|---|
| Pleasurable | Convenience Stability | Use of leisure time [29–31] | Ripple effect | Operational cost savings of transportation [31] |
| | | Appropriate control conversion [29–32] | | Gains against job losses in transportation [31] |
| | | Comfort driving [29,31,32] | | Personal data protection [30,31] |
| Usable | Usability | Easy to learn [29–31] | Social phenomena in commercialization | Autonomous driving infrastructure [30–32] |
| | | Complementary to poor driving [29–31] | | Clearness of legal liability [29,31–33] |
| | | | | Offering mobility to people unable to drive [31,32] |
| Reliable | Safety Utility | Fuel savings [29–32] | Social advantage expectation | Driving ethics [29–33] |
| | | Travel time savings [30–32] | | |
| | | Appropriate purchase/maintenance cost [29–32] | | Restricting illegal driving [30–32] |
| | | Drowsy/fatigue accident prevention [31,32] | | |
| | | Reduced driving stress/fatigue [31,32] | | Mitigating traffic congestion [29–31] |
| Functional | Functionality | Malfunction prevention ability [29–32] | - | |
| | | Situational awareness ability [29–32] | | |

The potential acceptance elements were classified according to the relationship between the human need level (presented in Section 3.1) and the Kano category. They were grouped by similarity into potential acceptance elements and structured by division into practical acceptance and social acceptance, according to the user requirement step. In the case of practical acceptance, the ability to be aware of the surrounding vehicles and traffic signals and to prevent malfunctions were classified as functional steps; this is the basic level of user requirements, because such aspects are conceived of as essential functions of autonomous driving. The elements related to user safety, economic efficiency (e.g., maintenance cost and fuel efficiency), and the prevention of drowsy driving were classified under the reliable level. The elements related to the interaction between autonomous vehicles and drivers (e.g., learning requirements) were classified under the usable level. Finally, acceptance elements that could enhance driver satisfaction (e.g., leisure time and comfort driving) were classified under the pleasurable level, the highest user stage. In the case of social acceptance, it was assumed that no acceptance element corresponded to the functional step. The elements directly related to driver safety (e.g., mitigation of traffic congestion), restricted illegal driving (e.g., drunk or unlicensed driving), or driving ethics (e.g., the trolley problem) were classified under the reliable step. The elements directly related to drivers (e.g., infrastructure for autonomous vehicles and legal liability of autonomous vehicle accidents) were designated under the usable step, and elements related to the ripple effect after the commercialization of autonomous vehicles (e.g., privacy and social operation effects) were classified under the pleasurable step.

### 3.3. Questionnaire Data Collection & Analysis

The questionnaire was composed of two parts. The first part was a questionnaire to measure the basic human characteristics, such as the respondents' demographics/experiences with autonomous driving technology. The latter part included the Kano questions designed to identify the category of each acceptance element. A brief introduction was given before

starting the survey, for which the respondents had no prior knowledge or experience regarding autonomous driving technology. The introduction focused on the characteristics related to the levels of automation. A total of 42 functional/dysfunctional items were prepared as questions with regard to 21 practical acceptance and social acceptance elements. A total of 187 respondents participated in the study, and the classification results of the respondents with respect to their personal characteristics are shown in Table 4.

**Table 4.** Questionnaire investigating the demographic aspects of respondents.

| Human Factors | Frequency | Human Factors | Frequency |
|---|---|---|---|
| Gender | | Age | |
| Male | 136 | 20's | 63 |
| Female | 51 | 30's | 40 |
| | | 40's | 26 |
| Driving experience | | 50's | 35 |
| ~2 yrs | 40 | 60's~ | 23 |
| 2~5 yrs | 21 | | |
| 5~10 yrs | 40 | Autonomous driving experience | |
| 10~15 yrs | 10 | None | 155 |
| 15~20 yrs | 29 | Level 1, 2 | 19 |
| 20 yrs~ | 47 | Level 3~ | 13 |

With the collected set of survey data, the category of each element was identified for all respondents. The distribution of the elements in the Kano diagram was confirmed by visualizing the S.I. and D.I. scores on the Kano model.

## 4. Results

An analysis was conducted to confirm the distribution of the users for the overall acceptance elements. This was designed to confirm whether the classification of an acceptance element was possible through the five categories of the Kano model. The Kano model determines the category with the highest frequency among those stated by the survey participants as the representative category. Although the acceptance elements were classified as different attributes to certain expectations, it was confirmed that most of the acceptance elements were classified into the expected category. The classification results of the acceptance elements for all the respondents are summarized in Table 5.

The two elements in the functional category were expected to be Must-be attributes because the users must be easily accessible via the graphic–user or voice–users interfaces in the car, making this a requirement; however, all were classified as One-dimensional attributes. This seems to imply that people do not believe that autonomous driving technology is functionally mature at the moment. The Mpa and the Saa still need to be improved. This is expressed as a Better/Worse index in Figure 4 below. It is considered to be the result of the user's perception that, if functions, such as the Mpa and the Saa (which reduce the causes of accidents in existing cars), do not work properly, they may threaten the user's safety. In particular, the Mpais related to malfunction prevention and was classified as a One-dimensional attribute; however, when comparing the Better/Worse index, it was found to be located close to the Must-be attribute. As autonomous driving technology progresses to Levels 4 and 5, it is predicted that the functional elements for accident prevention will become Must-be attributes.

**Table 5.** Categorized results of acceptance elements.

| Levels | | Acceptance Element | A | O | M | I | R | Q | S.I. | D.I. | Category |
|---|---|---|---|---|---|---|---|---|---|---|---|
| Pleasurable | Practical | Use of leisure time (Ult) | 73 | 82 | 7 | 19 | 0 | 6 | 0.86 | −0.49 | A |
| | | Appropriate control conversion (Acc) | 93 | 29 | 12 | 42 | 1 | 10 | 0.69 | −0.23 | A |
| | | Comfort driving (Cd) | 106 | 24 | 6 | 36 | 5 | 10 | 0.76 | −0.17 | A |
| | Social | Operational cost savings of transportation (Ocs) | 76 | 41 | 24 | 35 | 5 | 6 | 0.66 | −0.37 | A |
| | | Gains against job losses in transportation (Jol) | 68 | 9 | 14 | 84 | 0 | 12 | 0.44 | −0.13 | I |
| | | Personal data protection (Pdp) | 34 | 9 | 38 | 72 | 6 | 28 | 0.28 | −0.31 | I |
| Usable | Practical | Easy to learn (El) | 55 | 32 | 28 | 52 | 9 | 11 | 0.52 | −0.36 | A |
| | | Complementary to poor driving (Cpd) | 71 | 54 | 15 | 41 | 2 | 4 | 0.69 | −0.38 | A |
| | Social | Autonomous driving infrastructure (Adi) | 67 | 17 | 10 | 60 | 11 | 22 | 0.55 | −0.18 | A |
| | | Clearness of legal liability (Cll) | 13 | 96 | 59 | 14 | 1 | 4 | 0.60 | −0.85 | O |
| | | Offering mobility to people unable to drive (Omp) | 60 | 84 | 24 | 14 | 1 | 4 | 0.79 | −0.59 | O |
| Reliable | Practical | Fuel savings (Fs) | 112 | 42 | 8 | 17 | 2 | 6 | 0.86 | −0.28 | A |
| | | Travel time savings (Tts) | 105 | 35 | 12 | 24 | 1 | 10 | 0.80 | −0.27 | A |
| | | Appropriate purchase/maintenance cost (Apm) | 49 | 43 | 32 | 54 | 1 | 8 | 0.52 | −0.42 | A |
| | | Drowsy/fatigue accident prevention (Dap) | 30 | 91 | 35 | 12 | 6 | 13 | 0.72 | −0.75 | O |
| | | Reduced driving stress/fatigue (Rsf) | 52 | 90 | 23 | 14 | 3 | 5 | 0.79 | −0.63 | O |
| | Social | Driving ethics (De) | 1 | 1 | 44 | 37 | 68 | 36 | 0.02 | −0.54 | M |
| | | Restricting illegal driving (Rid) | 13 | 49 | 57 | 33 | 20 | 15 | 0.41 | −0.70 | M |
| | | Mitigating traffic congestion (Mtc) | 59 | 89 | 13 | 14 | 1 | 11 | 0.85 | −0.58 | O |
| Functional | Practical | Malfunction prevention ability (Mpa) | 25 | 64 | 51 | 29 | 2 | 16 | 0.53 | −0.68 | O |
| | | Situational awareness ability (Saa) | 49 | 83 | 30 | 12 | 3 | 10 | 0.76 | −0.65 | O |

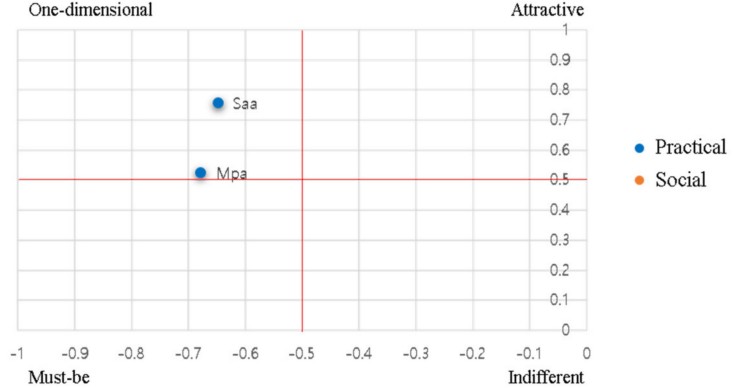

**Figure 4.** Factor positions for functional stage.

In terms of reliability, three elements (the Fs, Apm, and Tts) relating to time and cost were classified as Attractive attributes. These help to reduce the time and cost, but do not seem to cause much dissatisfaction (even in the current state) because of the conscious stereotypes regarding the time and costs involved with the existing car systems. In particular, the Apm is close to the Indifference attribute category, indicating that a significant number of people have not yet assessed the financial value of autonomous driving technology. This requires the establishment of a supply/demand model that considers the expected price to consumers because the acceptance may change significantly, depending on the consumer burden of the actual price of autonomous driving technology

in the future. In addition, the elements related to drowsiness/fatigue/stress appeared as One-dimensional attributes, suggesting that continuous development is required. In other words, if the autonomous driving technology advances to the 4th or 5th levels, and the user has enough technology to be "in-the-loop", the acceptance should be significantly affected. As expected, the factors related to driving ethics (related to the law under social acceptance and driving sanctions, such as those for unlicensed/drunk people) showed that regulation is required. In particular, clarifying the criteria of liability for accidents arising from "car without steering wheel" via national laws and policies will greatly contribute to increasing driver acceptance. The elements belonging to the reliable level are expressed as a Better/Worse index, as is shown in Figure 5 below.

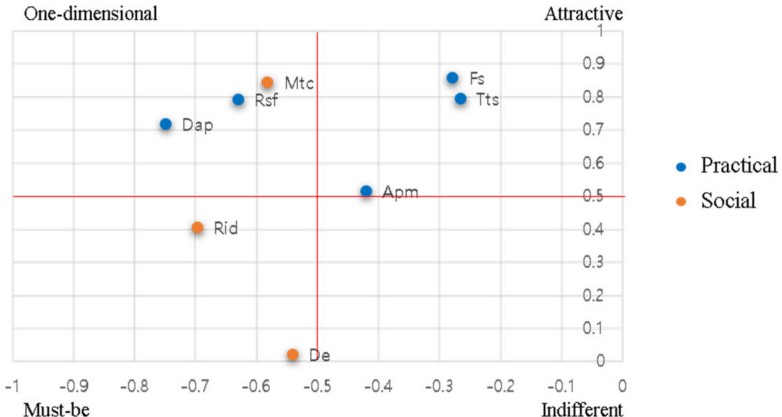

**Figure 5.** Factor positions for reliable stage.

In the usable stage, all the practical acceptance elements appeared as Attractive elements; among social acceptance, the Cll and Omp were confirmed as One-dimensional attributes. The Cll predicts that, in terms of the legal/ethical issues in the reliable level, clear regulations must be provided alongside the technology in order to secure acceptance. The Omp will require countermeasures that facilitate a sustainable society, where the disabled and people who cannot drive can live together with technology. "Easy to learn" is related to the learning ability and can be interpreted as follows: if the technology becomes easier to learn, greater acceptance can be secured. However, the degree is not large and is close to an Indifference attribute; thus, the current level of learning difficulty will not have a significant impact on acceptance. The Adi was also classified as an Attractive attribute but showed a similarity to the Indifference attributes. In other words, it does not have a significant effect on acceptance compared to the other elements because there are no specific application cases in the autonomous driving infrastructure, which is an essential element in Level-4 autonomous driving. When the level of autonomous driving technology reaches Levels 4 or 5, the importance of building traffic/road infrastructure (e.g., communication technology and traffic control) is expected to increase; accordingly, this aspect is expected to become a One-dimensional or Must-be attribute. Cpd relates to human error, showing that users are sufficiently satisfied with the current autonomous driving technology; it is judged that this will have a positive effect on securing acceptance when introducing advanced technology in the future. The elements belonging to the usable stage are expressed as a Better/Worse index in Figure 6 below.

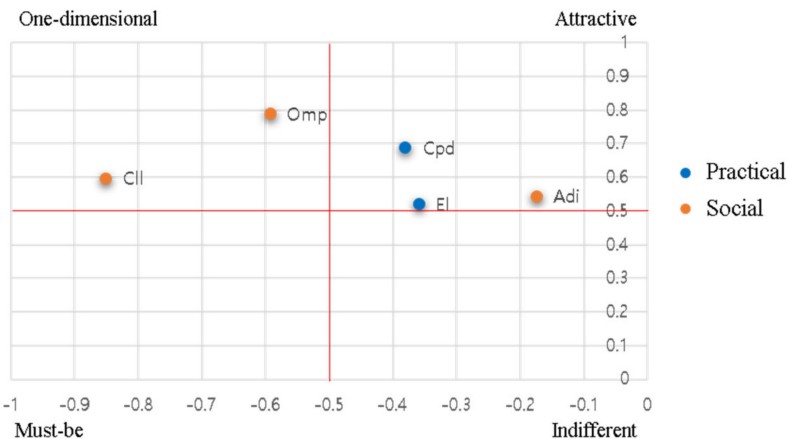

**Figure 6.** Factor positions for usable stage.

Most elements in the pleasurable stage appeared as Attractive attributes (as expected); however, in terms of the social acceptance elements, the Jol and Pdp appear as Indifference attributes, indicating that users have yet to consider what social impacts will arise in the future once autonomous driving technology has been commercialized. This may not have been a significant issue for the participants; accordingly, it seems to have been classified as an Indifferent attribute that is not yet a major problem. These acceptance elements will require reconfirmation as autonomous driving develops in the future. The pleasurable stage results, expressed as a Better/Worse index, are shown in Figure 7 below.

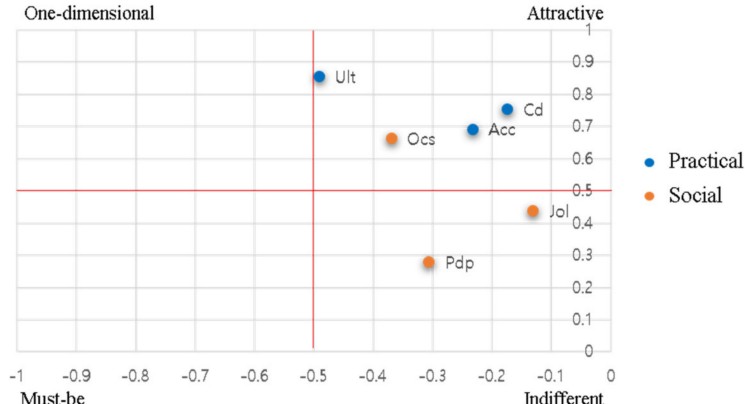

**Figure 7.** Factor positions for pleasurable stage.

Most of the acceptance elements correspond to the expected classification results, according to the user requirement level. It can be seen that the factors affecting acceptance are well classified; hence, the priority for improvement can be identified. Therefore, it is judged that if improvements are made by focusing on the elements belonging to the Must-be and One-dimensional attributes, which require continuous development (to secure the acceptance of the driver), greater acceptance will be secured.

## 5. Discussion

### 5.1. Kano-Model-Based Acceptance Classification Framework

In this study, the acceptance elements of autonomous driving technology were analyzed using a new method by applying the proposed acceptance research framework via the Kano model. The technology acceptance elements were derived by integrating the contents that have been suggested in various previous studies. In addition, by comparing the element classification criteria of the Kano model with the user requirement levels, we

identified the factors specifically related to the user requirements, as well as those that should be improved to secure acceptance.

In the study of acceptance, Davis's (1989) technical acceptance model (TAM) [34] is recognized as a very useful model for explaining the acceptor's information technology acceptance and behavioral intentions. The TAM evaluates the user acceptance through a questionnaire, and it has a measurement method similar to the Kano model. However, the early TAM has a drawback in that it cannot reflect the external factors affecting the technology acceptance process, and new models, such as the TAM2 and TAM3, have been developed to compensate for this disadvantage [35]. Yuen et al. (2021) used the TAM2 to confirm that several factors (e.g., recognition and compatibility) have a positive effect on the user's behavioral intentions for autonomous vehicles in terms of their perceived usefulness and perceived ease of use [36]. Hutchins et al. (2017) also used the TAM2 to examine the relationship between psychological factors, such as control, acceptance, and trust, for technology related to the safety of autonomous vehicles [37]. However, the TAM can only identify the correlation of factors affecting acceptance; it does not clearly distinguish the characteristics of each factor. It was confirmed that the Kano-model-based acceptance classification method presented in this study not only supplements the disadvantages of the TAM, but also confirms the changes in technology acceptance with respect to the various attributes for each acceptance element. In particular, the Kano model can compensate for the disadvantages of the TAM in that it can explore the priorities of the elements that have a sizeable influence on securing acceptance, according to the Better/Worse scores of the acceptance elements.

*5.2. Limitations and Future Research*

This study sought to identify the critical factors affecting the technological acceptance of autonomous driving, using the Kano model. In the process, it was verified that the model effectively confirmed the approximate strategies and directions required for securing the user acceptance of autonomous driving technology.

The findings of this study can potentially provide information for responding to acceptance changes in the future. However, this is macroscopic-level information because the research was conducted in a state where a significant number of potential factors were condensed. To establish a more specific strategy, additional research using detailed potential elements is needed. In addition, although this study collected the human characteristics of respondents with regard to the sociostatistical characteristics and the autonomous driving experience, we did not identify the differences in acceptance between the human characteristic groups. In the field of driver–autonomous car interactions, the human factors of the driver can exert a very large influence [38]. Therefore, further study of the relationship between human characteristics and the acceptance of autonomous driving technology is required.

In this study, a cross-sectional study was conducted to identify the elements determining the user acceptance of autonomous driving technology. However, human acceptance may vary over time. It is important to cross-check the changes in acceptance, though it is also important to evaluate and predict the changes in acceptability over time. Such longitudinal studies have already been conducted in other fields: Hu et al. (2003) conducted a longitudinal study of teachers [39] to evaluate the acceptance of information technology in the field of education; and Hogan et al. (2020) performed a longitudinal study on the adoption of new technologies for pharmacy staff in relation to robotic pharmacy dispensing systems [40]. Despite its status as a next-generation technology, longitudinal studies are lacking in the field of autonomous driving. In the future, it will be necessary to longitudinally evaluate the quality attributes for the subsequent changes in the user requirements. In particular, this study focused on the results for people with low levels of experience with regard to autonomous driving because most of the subjects lacked knowledge of the subject. However, many people will come to experience advanced autonomous driving technology; if users who have experienced autonomous driving are compared again, using

the acceptance elements derived from this study, a more realistic change in the acceptance may be confirmed. Tracking changes via the Kano model attributes over time could have a huge impact on the continued evolution of autonomous driving technology.

## 6. Conclusions

In this study, a new research framework was proposed and validated to complement the acceptance research (which is relatively inadequate in systematic research and analysis methods) for autonomous driving technology. The acceptance elements were defined according to the levels of the user requirements, and the priorities for improving acceptance were derived through element classification, using the Kano model. The users' perceptions of the malfunction prevention and situational awareness technology related to functionality (the lowest stages of user requirements) were expected to be Must-be attributes; however, it was confirmed that all can be considered One-dimensional attributes. This suggests that people are still aware that autonomous driving should be further developed, and, if it is further developed appropriately, a corresponding acceptance will be achieved. Among the acceptance elements belonging to the reliability level (predicted to appear as a Must-be or a One-dimensional attribute), the fuel economy, travel time, and purchase/maintenance costs were identified as Attractive attributes. It can be seen that no significant dissatisfaction was registered with the current state of autonomous driving technology in terms of these aspects, and they are judged to be subordinate priorities compared to the other elements. If driving ethics or restricted illegal driving (identified as Must-be attributes) are not satisfied, acceptance may significantly decrease. These factors are related to disadvantages in manual driving (e.g., drunk driving, reckless driving, etc.) and should be considered first in order to secure the acceptance of autonomous driving technology. Both the usable and pleasurable stages were classified as Attractive and Indifferent attributes. This is thought to be because autonomous driving technology is still in development, and the functional and reliable levels are not yet taken for granted by users. In the future, when all the lower-level requirements are satisfied, continued attention will be required, because this may change from an Attractive attribute to a One-dimensional attribute.

To conclude, if the acceptance elements belonging to the Must-be and One-dimensional attributes are first satisfied in the lower stages of the human requirements, greater acceptance can be secured. In addition, the results of this study are expected to contribute to the establishment of basic strategies for technological development and social countermeasures to facilitate the commercialization and continuous development of autonomous driving technologies. In this study, a new framework for quantitatively evaluating acceptance was presented, and it—together with the TAM—will serve as useful methodologies for acceptance evaluation studies in the future.

**Author Contributions:** Conceptualization, J.-H.Y. and S.-H.K.; methodology, J.-H.Y. and S.-H.K.; software, J.-G.S. and J.-H.Y.; validation, J.-G.S. and S.-H.K.; formal analysis, J.-G.S.; investigation, J.-H.Y.; resources, J.-H.Y., I.-S.H. and J.-G.S.; data curation, J.-G.S. and I.-S.H.; writing—original draft preparation, J.-G.S., J.-H.Y. and I.-S.H.; writing—review and editing, J.-G.S., I.-S.H. and S.-H.K.; visualization, J.-G.S. and I.-S.H.; supervision, S.-H.K.; project administration, S.-H.K.; funding acquisition, S.-H.K. All authors have read and agreed to the published version of the manuscript.

**Funding:** This research was funded by the Kumoh National Institute of Technology: 2018-104-125.

**Institutional Review Board Statement:** Ethical review and approval were waived for this study because this is a questionnaire study without experiments.

**Informed Consent Statement:** Informed consent was obtained from all the subjects involved in the study.

**Data Availability Statement:** The data presented in this study are available upon request from the corresponding author. The data are not publicly available because of privacy regulations.

**Conflicts of Interest:** The authors declare no conflict of interest.

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
