# Peer review of "How to Improve the Acceptance of Autonomous Driving Technology: Effective Elements Identified on the Basis of the Kano Model"

_applsci, doi:10.3390/app12031541_

Round 1

Reviewer 1 Report

+ Summary +

Authors review human and vehicle factors influencing user acceptance of AV papers and apply Kano model to classify and validate those factors

+ Strengths +

Important problem

Survey can be used to replicate past findings

Good review of relevant past literature

+ Weaknesses +

Method of how acceptance elements are derived is not sufficiently reproducible; specific analysis methods for generating the Kano metrics and figures is unclear

Method of determining human factors influencing acceptance is not reproducible

Authors claim that “specific definitions of the acceptance factors for autonomous driving…are also unclear” (page 2), but their definitions are not much more specific – for example, “situational awareness” (page 6) is not a specific design choice

Hypotheses can be more clearly worded

Author Response

Thanks to your advice, the manuscript has been significantly revised.

Please see the attached document for details.

happy New Year.

Reviewer 2 Report

This paper present and validate a research framework based on the Kano model to identify the effective acceptance factors for the autonomous driving technology.

While the paper is well-structured and anchored in previous literature, several parts are unclear and lack detailed explanations, making the scientific contribution unclear.

The use of a questionnaire to explore acceptance factors is valuable but is not sufficient; questionnaires may help us identify preferences and users’ perceptions, but they couldn’t measure absolute acceptance. The authors should consider defining their contributions better. 

The authors analyzed data of 187 respondents, differ by varioter their contribution. us demographic aspects; it would be interesting to present the results using cluster analysis. This can contribute to a user-centered design of future systems and may help understand better the results.

The authors should consider replacing the title of section 4. Results – this section is dealing with defining the different factors and preparing the questionnaire.  

Author Response

(The authors gave the same response as above.)
